# Lanostane Triterpenoids and Ergostane Steroids from *Ganoderma luteomarginatum* and Their Cytotoxicity

**DOI:** 10.3390/molecules27206989

**Published:** 2022-10-18

**Authors:** Qingyun Ma, Shuangshuang Zhang, Li Yang, Qingyi Xie, Haofu Dai, Zhifang Yu, Youxing Zhao

**Affiliations:** 1Haikou Key Laboratory for Research and Utilization of Tropical Natural Products, Institute of Tropical Bioscience and Biotechnology, Chinese Academy of Tropical Agricultural Sciences & Hainan Key Laboratory for Protection and Utilization of Tropical Bioresources, Hainan Academy of Tropical Agricultural Resource, Haikou 571101, China; 2Jiangsu Food&Pharmaceutical Science College, Huaian 223003, China; 3College of Food Science and Technology, Nanjing Agricultural University, Nanjing 210095, China

**Keywords:** *Ganoderma luteomarginatum*, lanostane triterpenoids, ergostane steroids, cytotoxic activity

## Abstract

Macrofungus *Ganoderma luteomarginatum* is one of the main species of *Ganoderma* fungi distributed in Hainan province of China, the fruiting bodies of which have been widely used in folk as a healthy food to prevent tumors. To explore the potential cytotoxic constituents from *G*. *luteomarginatum*, the phytochemical investigation on the ethyl acetate soluble fraction of 95% ethanolic extract from the fruiting bodies of this fungus led to the isolation of twenty-six lanostane triterpenoids (**1**–**26**), including three undescribed ones (**1**–**3**), together with eight ergostane steroids (**27**–**34**). The structures of three new lanostane triterpenoids were elucidated as lanosta-7,9(11)-dien-3β-acetyloxy-24,25-diol (**1**), lanosta-7,9(11)-dien-3-oxo-24,26-diol-25-methoxy (**2**), and lanosta-8,20(22)-dien-3,11,23-trioxo-7β,15β-diol-26-oic acid methyl ester (**3**) by the analysis of 1D, 2D NMR, and HRESIMS spectroscopic data. All isolates were assayed for their cytotoxic activities using three human cancer cell lines (K562, BEL-7402, and SGC-7901) and seven lanostane triterpenoids (**1**, **2**, **7**, **13**, **18**, **22**, and **24**), and one ergostane steroid (**34**) showed definite cytotoxicity with IC_50_ values that ranged from 6.64 to 47.63 μg/mL. Among these cytotoxic lanostane triterpenoids, compounds **2** and **1****3** showed general cytotoxicity against three human cancer cell lines, while compounds **1** and **18** exhibited significant selective cytotoxicity against K562 cells with IC_50_ values of 8.59 and 8.82 μg/mL, respectively. Furthermore, the preliminary structure–cytotoxicity relationships was proposed.

## 1. Introduction

The genus *Ganoderma*, belonging to the family Ganodermataceae and known as “Lingzhi” in Chinese, has been widely used as traditional Chinese medicine and functional foods for health in China and Southeast Asia for thousands of years [1]. There are more than 100 species in this family growing on cut or rotten trees in China, and 78 wild ones were found in Hainan Province [2]. As the major genus in this family, *Ganoderma* is a prolific producer of novel natural products responsible for its health benefits, mainly containing polysaccharides with an immunostimulative effect and triterpenes with a cytotoxic action [3]. Two main species, *G. lucidum* and *G. sinensis,* are recorded in Pharmacopoeia of the People’s Republic of China and used as an addition to conventional therapy in a clinical treatment of chronic bronchitis, bronchial asthma, leukopenia, coronary heart disease, arrhythmia, and acute infectious hepatitis. Recent research on chemical constituents of *Ganoderma* species showed that lanostane-type triterpenoids are the main characteristic natural products [4], and more than 400 lanostanoids have been isolated from the fungi of *Ganoderma*. These small molecule compounds have attracted considerable attention due to their extensive biological and pharmacological activities [5,6], including cytotoxic [7,8,9], hepatoprotective [10,11], anti-inflammatory [12,13,14], antidiabetic [15,16], neuroprotective [17], antiviral [18], antiaging [19], and antioxidant [20,21,22] effects. The genus *Ganoderma* is used as a healthy food and has been traditionally used for the prevention of numerous diseases or various pathological conditions, including complementary cancer therapy, especially a broad-spectrum application for the treatment of cancer.

Cancer has been considered as a huge threat to human health, and most governments are committed to diminishing this threat. The prevention and treatment of cancer becomes a key health goal. Finding antitumor drugs with high efficiency and low toxicity has become the urgent task, and countless researchers are dedicated to discovering bioactive ingredients from nature resources. *Ganoderma* is a promising anticancer immunotherapy agent owing to its low toxicology and efficacy as a combination therapy through the regulation of the immune system [23]. Polysaccharides and triterpenes from *Ganoderma* have been known to possess chemopreventive and antitumor activity. Many studies indicate that lanostane-type triterpenoids act as an inhibitor on different cancer cell lines, including the lung, liver, colon, pancreas, breast, skin, and prostate [6]. Among the reported active lanostanoids, the ganoderic acids are the main types of triterpene that play key roles in the biological activity. Lanostane-type triterpenoids can cause cell cycle arrest by the downregulation of cyclin D1 in the G1 phase of cell growth and inhibition of PKC activity in the G2 growth phase. Moreover, lanostane-type triterpenoids also prevent tumor metastasis by modulating MMPs and IL-8 and inhibit the excretion of inflammatory cytokines [24].

*Ganoderma luteomarginatum*, used as folk medicinal *Ganoderma* species, is a rare species mainly distributed in Hainan Province in China [25], where a pharmacodynamic molecular basis has been brought into focus in recent years [26,27]. In our ongoing endeavor to explore bioactive natural products, several species of *Ganoderma* have been studied, and a series of active compounds have been found [7,15,28,29,30,31]. The fruiting bodies of *G. luteomarginatum* have been widely used as a healthy food to prevent tumors. To explore the potential cytotoxic constituents from *G**. luteomarginatum*, we performed a phytochemical investigation on this fungus, which resulted in the isolation of twenty-six lanostane triterpenoids (**1**–**26**) (Figure 1), including three undescribed ones: lanosta-7,9(11)-dien-3β-acetyloxy-24,25-diol (**1**), lanosta-7,9(11)-dien-3-oxo-24,26-diol-25- methoxy (**2**), and lanosta-8,20(22)-dien-3,11,23-trioxo-7β,15β-diol- 26-oic acid methyl ester (**3**), together with eight ergostane steroids (**27**–**34**) (Figure 1). All these isolates were evaluated for their cytotoxic activity against three human cancer cell lines. Herein, we reported the isolation, structural elucidation, and cytotoxicity of the compounds isolated from the fruiting bodies of *G. luteomarginatum*.

## 2. Results and Discussion

### 2.1. Structural Elucidation of Compounds

Compound **1** was obtained as white amorphous powder, and its molecular formula was determined to be C_32_H_52_O_4_ on the basis of HRESIMS ion at *m/z* [M + Na]^+^ 523.3754 (calcd. 523.3763 for C_3__2_H_52_NaO_4_^+^), indicating seven degrees of unsaturation. The IR spectrum revealed the presence of hydroxyl groups (3475 cm^−1^), double bonds (1641 cm^−1^), and ester carbonyl (1696 cm^−1^). The ^1^H NMR spectral data (Table 1) of **1** revealed the signals for nine methyls (*δ*_H_ 0.58; 0.89; 0.90; 0.93, d, *J* = 6.3 Hz; 0.97; 1.02; 1.18; 1.23; 2.07), one proton related to oxygenated carbon (*δ*_H_ 4.53, dd, *J* = 11.4, 4.6 Hz), and two olefinic protons (*δ*_H_ 5.47, t, *J* = 4.5 Hz; 5.34, d, *J* = 5.9 Hz). The ^13^C NMR and DEPT (Table 1) spectra presented 32 carbon signals for nine methyls; eight methylenes; seven methines (two oxygenated and two olefinic); and eight non-protonated carbons (two olefinic, one oxygenated, and one ester carbonyl at *δ*_C_ 171.2). The above-mentioned NMR data were closely similar to those of lanosta-7,9(11)-dien-3b-acetyloxy-24,25,26-trihydroxy [29], suggesting that **1** had a lanostane skeleton and structurally similar to this compound. The only difference between them was that the methylol group (*δ*_C_ 67.8) at C-26 in lanosta-7,9(11)-dien-3b-acetyloxy-24,25,26-trihydroxy was replaced by one methyl (*δ*_C_ 25.7) in **1**, which was confirmed by HMBC correlations (Figure 2) from H_3_-27 (*δ*_H_ 1.18) to C-26, C-24 (*δ*_C_ 79.7) and C-25 (*δ*_C_ 73.4). The attachment of acetate group to C-3 was proposed by the key HMBC correlation of H-3 (*δ*_H_ 4.53) with acetal carbonyl (*δ*_C_ 171.2). The other obvious HMBC correlations (Figure 2) of **1** from H_3_-28 (*δ*_H_ 0.90) and H_3_-29 (*δ*_H_ 0.97) to C-3, C-4 (*δ*_C_ 37.9), and C-5 (*δ*_C_ 49.3); from H_3_-18 (*δ*_H_ 0.55) to C-12 (*δ*_C_ 37.7), C-14 (*δ*_C_ 50.4), and C-17 (*δ*_C_ 51.1); from H_3_-19 (*δ*_H_ 1.02) to C-1 (*δ*_C_ 35.5), C-5, and C-9 (*δ*_C_ 145.7); and from H_3_-30 (*δ*_H_ 0.89) to C-8 (*δ*_C_ 142.8), C-13 (*δ*_C_ 43.8) and C-15 (*δ*_C_ 28.0) further assigned its planar structure of lanostanoid. The relative configuration of the tetracyclic core structure of **1** was determined to be the same as that of lanosta-7,9(11)-dien-3b-acetyloxy- 24,25,26-trihydroxy by comparison of their NMR and ROESY spectroscopic data (Figure 1), revealing ROESY correlations of H-18 with H-19 and H-20 (*δ*_H_ 1.39). The key ROESY correlation of H-3 (*δ*_H_ 4.53) with H_3_-28 (*δ*_H_ 0.90) and H-5 (*δ*_H_ 1.18) suggested the assignment of their same orientation. Based on comprehensive analysis of 1D, 2D NMR, and HRESIMS spectrums (see Appendix A), the structure of compound **1** was elucidated to be lanosta-7,9(11)-dien-3β-acetyloxy-24,25-diol.

Compound **2** had the molecular formula C_31_H_50_O_4_, as determined by the HRESIMS ion peak at 509.3597 (calcd. 509.3607 for C_31_H_50_NaO_4_^+^). The ^13^C NMR and DEPT (Table 1) spectra showed 31 carbon signals for eight methyls (one methoxy at *δ*_C_ 49.5); nine methylenes (one oxygenated at *δ*_C_ 64.4); six methines (one oxygenated at *δ*_C_ 76.8 and two olefinic); and eight non-protonated carbons (two olefinic, one oxygenated, and one ketone carbonyl at (*δ*_C_ 217.1). The NMR spectra of **2** resembled those of (24*S*,25*R*)-25- methoxylanosta-7,9(11)-dien-3β,24,26-triol [32], except for the presence of a ketone carbonyl of C-3 (*δ*_C_ 217.1) in **2** replacing the hydroxylated methine in (24*S*,25*R*)-25- methoxylanosta-7,9(11)-dien-3β,24,26-triol, which was corroborated by the HMBC correlations (Figure 2) from H_3_-28 (*δ*_H_ 1.10), H_3_-29 (*δ*_H_ 1.14), and H_2_-2 (*δ*_H_ 2.38/2.80) to C-3 (*δ*_C_ 217.1). The attachment of methoxy to C-25 was assigned by the key HMBC correlation of the protons signal at *δ*_H_ 3.34 with C-25 (*δ*_C_ 78.5). The other clear HMBC correlations (Figure 2) of **1** from H_3_-28 (*δ*_H_ 1.10), H_3_-29 (*δ*_H_ 1.14), and H_3_-19 (*δ*_H_ 1.21) to C-5 (*δ*_C_ 50.8) and from H_3_-18 (*δ*_H_ 0.61) and H_3_-30 (*δ*_H_ 0.89) to C-13 (*δ*_C_ 43.9) and C-14 (*δ*_C_ 50.4) further confirmed its planar structure of lanostanoid. The relative configuration of **2** was established as same as that of (24*S*, 25*R*)-25-methoxylanosta-7,9(11)-dien-3β,24,26-triol by the ROESY spectrum (Figure 2), revealing key ROESY correlations of H-30 with H-17 (*δ*_H_ 1.61) and of H-18 with H-19 and H-20 (*δ*_H_ 1.44), as well as their similar NMR data. On the basis of the above evidence, the structure of **2** was defined as lanosta-7,9(11)-dien- 3-oxo-24,26-diol-25-methoxy.

The HRESIMS ion peak 529.3160 [M+H]^+^) of compound **3** gave the molecular formula C_31_H_44_O_7_, indicating ten degrees of unsaturation. The ^13^C NMR and DEPT (Table 1) spectra showed 31 carbon signals for eight methyls (one methoxy at *δ*_C_ 52.0), six methylenes, six methines (two oxygenated, and one olefinic at *δ*_C_ 124.7) and eleven non-protonated carbons (three olefinic; three ketones at *δ*_C_ 216.8, 199.4, and 198.4; and one ester carbonyl *δ*_C_ 176.7). A detailed analysis of the 1D NMR data of **3** suggested that compound **3** possessed the same planar structure as methyl ganoderenate A [33] with a lanostane skeleton. The whole connectivity of compound **3** was also further demonstrated by 2D NMR data, including HMBC spectrum (Figure 2), exhibiting HMBC correlations from H_3_-28 (*δ*_H_ 1.13), H_3_-29 (*δ*_H_ 1.12), and H_3_-19 (*δ*_H_ 1.36) to C-5 (*δ*_C_ 49.3) and from H_3_-18 (*δ*_H_ 1.10) and H_3_-30 (*δ*_H_ 1.24) to C-13 (*δ*_C_ 48.5) and C-14 (*δ*_C_ 57.1). The relative configuration of the tetracyclic core structure of **1** was determined to be the same as methyl ganoderenate A, except for chiral C-15 by analysis of its ROESY spectrum (Figure 2). The key ROESY correlations of H-7 (*δ*_H_ 4.80) with H_3_-30 (*δ*_H_ 1.24) and H-5 (*δ*_H_ 1.67) proposed the β-orientation of hydroxy at C-7. The β-orientation of hydroxy at C-15 was assigned by the key ROESY correlation of H_3_-30 (*δ*_H_ 1.24) with H-15 (*δ*_H_ 4.33). Moreover, the key ROESY correlation of H-22 (*δ*_H_ 6.16) with H-17 (*δ*_H_ 2.76) indicated that the geometry of double bond *∆*^20(22)^ in **3** was *E* instead of *Z* in methyl ganoderenate A. Therefore, the structure of compound **3** was established as lanosta-8,20(22)-dien-3,11,23-trioxo-7β,15β- diol- 26-oic acid methyl ester.

The thirty-one known compounds, including twenty-one lanostane triterpenoids (**4**–**26**) and eight ergostane steroid (**27**–**34**), were identified as lanosta-8,24*E*-dien-7-oxo-3β- acetyloxy-26-ol (**4**) [26], lanosta-8,24*E*-dien-7-oxo-3β-acetyloxy-26-al (**5**) [26], lanosta-7,9(11),24-trien-3β-acetyloxy-26-ol (**6**) [26], (24*R*,25*S*)-lanosta-7,9(11)-dien- 3β,24,26-triol-25-methoxy (**7**) [16], lanosta-7,9(11)-dien-3β-acetyloxy-24,25,26-triol (**8**) [29], lanosta-7,9(11)-dien-3β-acetyloxy-24,26-dihydroxy-25-methoxy (**9**) [29], ganodermanondiol (**10**) [34], lucidumol B (**11**) [34], 26-hydroxy-ganodermanondiol (**12**) [35], ganoderiol A (**13**) [36], ganoderone A (**14**) [37], lucidadiol (**15**) [38], ganodermadiol (**16**) [39], lanosta-7,9(11),24*E*-trien-3β-acetyloxy-26,27-diol (**17**) [29], ganoderiol F (**18**) [40], lanosta-8-en-7,11-dioxo-3β-acetyloxy-24,25,26-triol (**19**) [29], ganoderiol D (**20**) [40], lanosta-8-en-7-oxo-3β-acetyloxy-24,25,26-trihydroxy (**21**) [34], lucidumol A (**22**) [41], lanosta-7,9(11),24*E*-trien-3-oxo-26-al (**23**) [42], lanosta-7,9(11),24-triene-3β-ol-26-al (**24**) [42], lucidone H (**25**) [29], lucidadone H (**26**) [43], ergosta-7,9(11),22*E*-triene-3β,5*α*-diol -6β-methoxy (**27**) [29], ergosta-7,22*E*-dien-3-one (**28**) [44], ergosta-7,22*E*-dien-3β-ol (**29**) [45], ergosta-4,6,8(14),22*E*-tetraen-3-one (**30**) [44], ergosterol peroxide (**31**) [46], ergosta-3β,5*α*-diol-7,22*E*-dien-6β-methoxy (**32**) [47], ergosta-7,22*E*-dien-3β,5*α*,6β-triol (**33**) [48], and calvasterol B (**34**) [49] by comparing their NMR data with those reported in the literature.

*Ganoderma* has been used as a healthy food and medicinal purposes for centuries particularly in China, Japan, and Korea. A great deal of work has been carried out on over thirty species of *Ganoderma*. Two types of natural products, lanostane triterpenoids (**1**–**26**) and ergostane steroids (**27**–**34**), were discovered from the fruiting bodies of *G. luteomarginatum* collected in Hainan Province, China. These two types of compounds are widely found in genus *Ganoderma* [1], and lanostanoids (*Ganoderma* triterpenoids) was the characteristic active metabolites in species of *Ganoderma**,* which are a class of compounds with various chemical structures. Here, the isolated lanostane triterpenoids were divided into two groups according to the conjugated system at C-7, C-8, C-9, and C-11. The first group possessed the conjugated double bond *∆*^7,9^^(^^11^^)^ as shown in **1**, **2**, **6**–**18**, **23,** and **24**. The second group had (*∆*^8^)α,β-unsaturated ketone at C-7 or C-9. The C-26 in lanostanoid structures is often oxidized to alcohols, aldehydes, and acids [5]. Among the structures of identified lanostanoids (**1**–**26**), ganoderiol derivative was the main type. In addition, three ganoderic aldehydes (**5**, **23**, **24**) and one ganoderic acid (**3**) were also found. The *β*-configuration of OH-15 in new ganoderic acid (**3**) from *G. luteomarginatum* was consistent with the previously discovered ganoderic acid derivatives from this fungus [27], which was opposite of that shown in the corresponding compounds from other *Ganoderma* species [7,12,50]. Moreover, norlanostanoids with 24 carbon atoms often occur in *Ganoderma*. In present study, two hexanorlanostanoids (**25** and **26**) were isolated. Some lanostane triterpenoids besides the three new ones in our study are structurally different from those previously reported lanostanoids [26,27] from *G. luteomarginatum* collected in Guangxi Province, China. This may be related to the different growth environment of this fungus or different growth period for collection, which needs further comparative analysis in our subsequent studies.

### 2.2. Cytotoxic Activities of Compounds

The cytotoxic activities of all the isolates were evaluated by MTT method toward three human cancer cell lines (K562, BEL-7402, and SGC-7901). The results were presented in Table 2. Of the compounds tested, seven lanostane triterpenoids (**1**, **2**, **7**, **13**, **18**, **22**, and **24**) showed definite cytotoxicity against K562 with IC_50_ values range from 6.64 to 17.38 μg/mL, among which compounds **1**, **13** and **18** showed the IC_50_ values of 8.59, 6.64, and 8.82 μg/mL, respectively. Compounds **2** and **13** also showed moderate cytotoxicity against two human cancer cell lines (BEL-7402 and SGC-7901). Moreover, compound **7** showed moderate cytotoxicity against human cancer cell lines BEL-7402 with IC_50_ value of 20.05 μg/mL. Compounds **1**, **18**, **22,** and **24** had no obvious cytotoxicity on BEL-7402 and SGC-7901 cell lines (IC_50_ > 50 μg/mL). Among these cytotoxic lanostanoids, compounds **2** and **1****3** showed general cytotoxicity against three human cancer cell lines, while compounds **1** and **18** exhibited significant selective cytotoxicity against K562 cell lines. One ergostane steroid (**34**) showed general cytotoxicity against three human cancer cell lines.

Lanostane-type triterpenoids of *Ganoderma* are considered to be the major pharmacologically active compounds that contribute to its antitumor efficacy. The lanostane-type triterpenoids were extensively evaluated for cytotoxic activities against a series of tumor cell lines [5] related to lung, liver, colon, pancreas, breast, skin, and prostate [6]. The lanostanoids with structural complexity and functional group variety may be specific to different cell lines and the structure-cytotoxicity relationships could be raised. Compounds **1**, **18**, **22,** and **24** exhibited selective cytotoxicity against K562 cell lines may be due to their unique structures. From the results of isolated lanostanoids against three human cancer cell lines (K562, BEL-7402, and SGC-7901), the conjugated double-bond *∆*^7,9(^^11)^ system in tetracyclic skeleton (**1**, **2**, **7**, **13**, **18**, and **24**) seemed to be more important than (*∆*^8^)α,β-unsaturated ketone system for potent cytotoxic activity. Comparing the cytotoxicity between **7**/**9**, **18**/**17,** and **24**/**5** with only difference at C-3, it suggested that acetylation may be the negative factor for cytotoxic activity. In addition, compound **13** exhibited significant cytotoxicity, while their keto-3 analog **12** was inactive, assumed that reduction of the keto-3 group to OH-3 in lanostane triterpenoids would improve the cytotoxicity against K562, BEL-7402, and SGC-7901 significantly. The above preliminary structure–cytotoxicity relationships provide an approach to understanding the structural requirements of lanostane-type triterpenoids.

## 3. Materials and Methods

### 3.1. General Experimental Procedures

The NMR spectra were recorded with a Bruker AV-500 spectrometer (Bruker, Bremen, Germany) with TMS as an internal standard. HRESIMS data were determined on a mass spectrometer API QSTAR Pulsar (Bruker, Bremen, Germany). Optical rotations were measured on a Rudolph Autopol III polarimeter (USA). UV spectra were obtained on a Shimadzu UV-2550 spectrometer. IR spectra were obtained on a Nicolet 380 FT-IR spectrometer with KBr pellets. Silica gel (60–80 and 200–300 mesh, Marine Chemical Industry Factory, Qingdao, China), Rp-C18 (20–45 mL; Fuji Silysia Chemical Ltd., Aichi, Japan), and Sephadex LH-20 (Merck, Germany) were used for column chromatography. Fractions were monitored by TLC and spots were visualized by heating after spraying with 5% H_2_SO_4_ in ethanol.

### 3.2. Fungal Material

Fruiting bodies of G. luteomarginatum were collected in Qiongzhong County, Hainan Province, China (June, 2012), and identified by Prof. Xing-Liang Wu of Hainan University. A voucher specimen (No. 2012HB01) is deposited at the Institute of Tropical Bioscience and Biotechnology, Chinese Academy of Tropical Agricultural Sciences.

### 3.3. Isolation and Characterization of Compounds

Dried and powdered fruiting bodies of G. luteomarginatum (2.5 kg) were extracted with EtOH-H_2_O (10 L, 95:5, *v*/*v*) under reflux conditions three times at a duration of 4 h. The combined extracts were concentrated and suspended in H_2_O, followed by successive partitioning with EtOAc and n-BuOH, respectively. The EtOAc extract (53.0 g) was separated by silica gel column chromatography (CC) under reduced pressure using a solvent gradient of petroleum ether (PE)-EtOAc (20:1→0:1, *v*/*v*) to afford six fractions (Fr1-Fr6). Fr2 (6.2 g) was subjected to silica gel CC under reduced pressure eluted with PE-EtOAc (3:1) to give 3 subfractions: 2a–2c. Subfraction 2a (925 mg) was repeatedly purified by silica gel CC eluted with PE-EtOAc (5:1) to obtain compounds **4** (7.3 mg), **5** (4.2 mg), **6** (5.2 mg), **14** (21.5 mg), **26** (28.7 mg), and **30** (16.4 mg). Subfraction 2b (816 mg) was separated by silica gel column to yield compounds **23** (7.7 mg), **24** (6.0 mg), and **28** (68.3 mg) using an eluent CHCl_3_-MeOH (15:1), and compounds **25** (2.4 mg) and **29** (81.7mg) were separated from subfraction 2c (674 mg) using an eluent: CHCl_3_-MeOH (12:1). Fr3 (9.6 g) was separated by Rp-18 CC with MeOH-H_2_O (30:70→0:100) to give 3 subfractions: 3a–3c. Subfraction 3a (526 mg) was purified by silica gel CC eluted with PE-EtOAc (3:1) to obtain compounds **1** (5.5 mg), **2** (8.2 mg), **9** (8.6 mg), and **10** (9.0 mg). Subfraction 3b (603 mg) was separated by silica gel CC eluted with PE-EtOAc (2:1) to yield compounds **3** (4.6 mg), **15** (5.5 mg), **16** (5.1 mg), **22** (10.5 mg), and **31** (27.0 mg). Subfraction 3c (839 mg) was subjected to silica gel CC eluted with CHCl_3_-MeOH (10:1) to yield compounds **17** (8.6 mg), **18** (9.8 mg), **27** (3.9 mg), **32** (3.8 mg), and **34** (5.1 mg). Fr4 (8.5 g) was treated by Rp-18 CC with MeOH-H_2_O (30:70→0:100) to afford subfractions 4a–4d. Subfraction 4b (406 mg) was subjected to Sephadex LH-20 (CHCl_3_/MeOH, 1:1), then by silica gel CC with PE-EtOAc (2:1) to yield compounds **7** (8.1 mg), **8** (7.6 mg), and **19** (7.4 mg). Compounds **11** (6.5 mg) and **12** (15.6 mg) was purified from subfraction 4b (365 mg) using an eluent CHCl_3_-MeOH (8:1). Fr5 (9.0 g) was separated by Rp-18 CC with MeOH-H_2_O (30:70→0:100) to afford subfractions 5a–5c. Subfraction 5b (582 mg) was subjected to silica gel CC with CHCl_3_-EtOAc (2:1) to yield compounds **13** (7.8 mg) and **20** (10.2 mg). Subfraction 5c (264 mg) was separated by silica gel CC eluted with CHCl_3_-MeOH (6:1) to obtain **21** (8.4 mg) and **33** (25.8 mg).

Lanosta-7,9(11)-dien-3β-acetyloxy-24,25-diol (**1**): White amorphous powder; [α]^27^_D_ −2.5° (c 0.02, MeOH); UV (MeOH) λ_max_ (logε) 252 (3.62), 242 (5.24), 238 (4.50), 210 (1.21); IR (KBr) ν_max_ cm^−1^ 3475, 2938, 1696, 1641, 1302, 1028; ^1^H and ^13^C NMR data; see Table 1; HRESIMS m/z [M+Na]^+^ 523.3754 (calcd. 523.3763 for C_32_H_52_NaO_4_).

Lanosta-7,9(11)-dien-3-oxo-24,26-diol-25-methoxy (**2**): White amorphous powder; [α]^27^_D_ +4.0° (c 0.02, MeOH); UV (MeOH) λ_max_ (log ε) 242 (5.60), 238 (5.03), 212 (2.32); IR (KBr) v_max_ cm^−1^ 3424, 2952, 1724, 1644, 1386, 1020; ^1^H and ^13^C NMR data; see Table 1; HRESIMS m/z [M+Na]^+^ 509.3597 (calcd. 509.3607 for C_31_H_50_NaO_4_).

Lanosta-8,20(22)-dien-3,11,23-trioxo-7β,15β-diol-26-oic acid methyl ester (**3**): White amorphous powder; [α]^27^_D_ −1.5° (c 0.02, MeOH); UV (MeOH) λ_max_ (log ε) 250 (3.54), 210 (1.37); IR (KBr) v_max_ cm^−1^ 3443, 2934, 1732, 1639, 1384, 1019; ^1^H and ^13^C NMR data; see Table 1; HRESIMS m/z 529.3160 [M+H]^+^ (calcd. 529.3165 for C_31_H_45_O_7_).

### 3.4. Bioassay of Cytotoxic Activity

All the compounds was assayed for their cytotoxic activity against three human tumor cell lines: K562 (leukemic cell line), BEL7402 (hepatoma cell line), and SGC7901 (gastric cancer cell line) using MTT methods reported previously [7]. Briefly, each tested compound was dissolved with DMSO at concentration of 10 mM and then diluted to the required concentrations with the medium. Cells were cultured in 96-well plates with initial density of 5000 cells/well 12 h before treatment and exposed to different concentrations (40, 8, 1.6, 0.32, and 0.064 μM, respectively) of compounds, with paclitaxel (Sigma, Livonia, MI, USA) as the positive control. After the culturing period, 20 μL of MTT (5 mg/mL) was added per well and incubated for 4 h at 37 °C. Finally, absorbance was measured at 570 nm using a microplate reader. Each assay was replicated three times. The effect of the compounds on cell viability was calculated and expressed as the IC_50_.

## 4. Conclusions

Lanostane-type triterpenoids are the main characteristic natural products of the fungi of *Ganoderma*, which have extensive biological and pharmacological activities, especially possess cytotoxicity. The chemical investigation of the fruiting bodies of *G*. *luteomarginatum* led to the isolation of twenty-six lanostane triterpenoids (**1**–**26**), including three undescribed ones (**1**–**3**): lanosta-7,9(11)-dien-3β-acetyloxy-24,25-diol (**1**), lanosta-7,9(11)-dien-3-oxo-24,26-diol-25-methoxy (**2**), and lanosta-8,20(22)-dien-3,11,23-trioxo -7β,15β-diol-26-oic acid methyl ester (**3**), together with eight ergostane steroids (**27**–**34**). The cytotoxicity assay showed that seven lanostane triterpenoids (**1**, **2**, **7**, **13**, **18**, **22**, and **24**) revealed definite cytotoxicity against K562, among which compounds **1**, **13,** and **18** showed the IC_50_ values of 8.59, 6.64, and 8.82 μg/mL, respectively, indicating the prospect of an antitumor. Some preliminary structure–cytotoxicity relationships of these lanostane triterpenoids showed that the conjugated double-bond *∆*^7,9^^(^^11^^)^ system in tetracyclic lanostane skeleton seemed to be more important than the (*∆*^8^)α,β-unsaturated ketone system for potential cytotoxic activity. The present study further enriched the understanding of the structural diversity of *G*. *luteomarginatum*, which also provides theoretical information for its subsequent anticancer drug development.

## Figures and Tables

**Figure 1 molecules-27-06989-f001:**
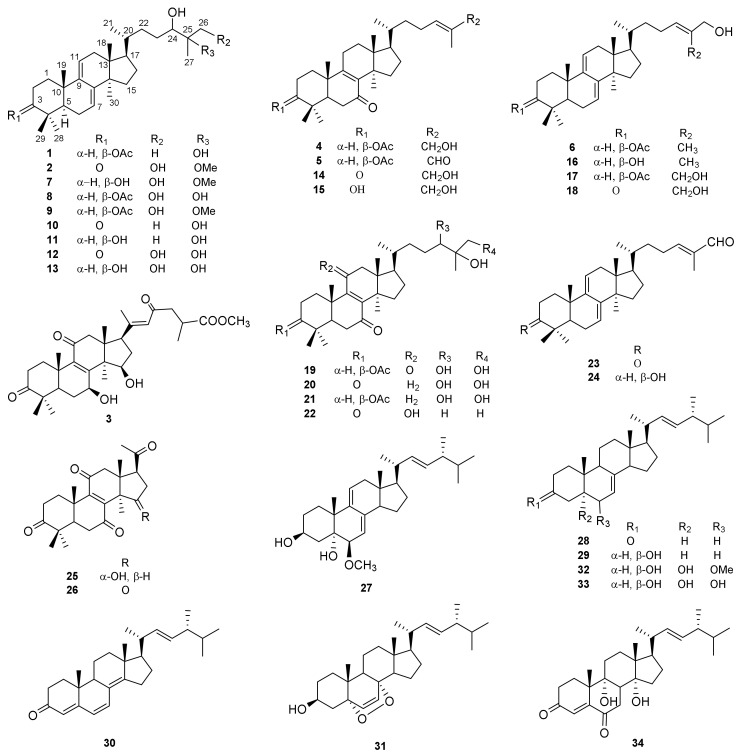
The structures of compounds **1**–**34**.

**Figure 2 molecules-27-06989-f002:**
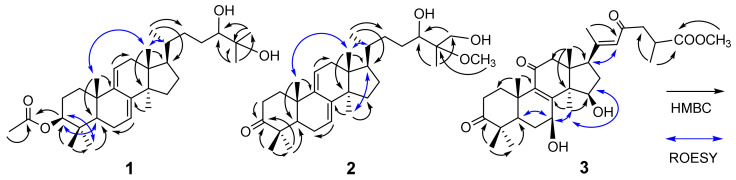
Key HMBC (H→C) and ROESY (↔) correlations of **1**–**3**.

**Table 1 molecules-27-06989-t001:** ^1^H (500 MHz) and ^13^C NMR (125 MHz) Data of Compounds **1**–**3** (in CDCl_3_).

No.		1		2		3
	*δ* _C_	*δ*_H_ (*J* in Hz)	*δ* _C_	*δ*_H_ (*J* in Hz)	*δ* _C_	*δ*_H_ (*J* in Hz)
1	35.5	1.97 m 1.50 m	36.7	2.36 m 1.77 m	35.6	2.45 m 2.06 m
2	24.4	1.71 m	35.0	2.80 dt (14.7,5.7) 2.38 ddd (14.7, 4.5, 3.1)	34.4	2.85 m 2.36 m
3	81.0	4.53 dd (4.6, 11.4)	217.1		216.8	
4	37.9		47.6		46.9	
5	49.3	1.18 m	50.8	1.56 dd (3.7, 11.9)	49.3	1.67 dd (13.5, 2.2)
6	22.9	2.08 m	23.8	2.02 m 2.23 m	36.5	2.11 m 1.84 m
7	120.1	5.47 t (4.5)	120.0	5.52 d (6.8)	69.5	4.80 dd (7.1, 9.4)
8	142.8		143.0		157.0	
9	145.7		144.6		143.3	
10	37.3		37.3		38.3	
11	116.6	5.34 d (5.9)	117.4	5.40 d (6.2)	199.4	
12	37.7	2.23 d (17.6) 2.02 m	37.9	2.28 m 2.08 m	51.7	2.75 d (14.6) 2.37 d (14.6)
13	43.8		43.9		48.5	
14	50.4		50.4		57.1	
15	28.0	1.29 m	28.0	1.27 m	77.8	4.33 d (6.7)
16	31.6	1.60 m 1.36 m	31.6	1.67 m 1.40 m	35.3	2.35 m 1.42 m
17	51.1	1.58 m	51.1	1.61 m	54.5	2.76 m
18	15.8	0.55 s	15.9	0.61 s	19.6	1.10 s
19	23.3	1.02 s	22.2	1.21 s	19.7	1.36 s
20	36.7	1.39 m	36.7	1.44 m	156.8	
21	18.7	0.93 d (6.3)	18.7	0.94 d (6.5)	21.3	2.16 s
22	33.6	1.76 m	33.8	1.86 m	124.7	6.16 s
23	28.8	1.56 m	28.4	1.63 m	198.4	
24	79.7	3.31 dd (10.2, 2.1)	76.8	3.59 d (9.4)	47.8	2.94 m 2.57 m
25	73.4		78.5		35.0	2.96 m
26	25.7	1.23 s	64.4	3.75 d (12.0) 3.65 d (12.0)	176.7	
27	22.9	1.18 s	16.0	1.05 s	17.3	1.20 d (6.8)
28	28.2	0.90 s	25.4	1.10 s	27.0	1.13 s
29	17.1	0.97 s	22.6	1.14 s	21.0	1.12 s
30	25.7	0.89 s	25.6	0.89 s	26.0	1.24 s
OAc	171.2 21.5	2.07 s				
OMe			49.5	3.34 s	52.0	3.69 s

**Table 2 molecules-27-06989-t002:** Cytotoxic activities of compounds from *Ganoderma luteomarginatum* (IC_50_, μg/mL).

Compounds	K562	BEL-7402	SGC-7901
**1**	8.59	>50	>50
**2**	16.05	24.27	33.38
**7**	11.69	20.05	>50
**13**	6.64	13.49	15.62
**18**	8.82	>50	>50
**22**	16.95	>50	>50
**24**	17.38	>50	>50
**34**	22.81	47.63	26.06
Paclitaxel	5.62	3.26	3.41

## Data Availability

Not applicable.

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
