# Peer review of "Lanostane Triterpenoids and Ergostane Steroids from Ganoderma luteomarginatum and Their Cytotoxicity"

_molecules, 2022, doi:10.3390/molecules27206989_

Round 1
Reviewer 1 Report
I reviewed the manuscript titled “Lanostane triterpenoids and ergostane steroids from the fruiting bodies of Ganoderma luteomarginatum and their cytotoxicity” for your esteemed ‘MDPI Molecules’. This study is not a novelty.
The authors investigated “Lanostane triterpenoids and ergostane steroids from the fruiting bodies of Ganoderma luteomarginatum and their cytotoxicity”.
Other comments
In view of this tight competitive situation, only manuscripts with highly innovative contents, originality, gain in scientific knowledge, straightforward experimental design, advanced state-of-art technical performance of experiments, and general interest for the readership can be considered.
In the present case, your manuscript represents there is no novelty for scientifically sound contribution because already published this kind of objective, please refer to below attached hyperlinks for your kind perusal. However, it did not reach a sufficiently high priority index concerning novelty and relevance to a broad scientific readership. Therefore, we feel that your work would not be accepted.
Decision:Reject
Links:
Ganoderma luteomarginatum - Search Results - PubMed (nih.gov)
https://scholar.google.com/scholar?hl=en&as_sdt=0%2C5&q=+ergostane+lanostane+Ganoderma+luteomarginatum&btnG=
Author Response
Thanks for your comments and suggestions very much. Ganoderma luteomarginatum is a rare species and used as a healthy food to prevent tumors in folk. This manuscript mainly describes the isolation of twenty six lanostane triterpenoids including three undescribed ones, together with eight ergostane steroids, from the fruiting bodies of Ganoderma luteomarginatum. Seven lanostane triterpenoids and one ergostane steroid showed definite cytotoxicity. The present study further enriched the understanding of structural diversity of G. luteomarginatum, which also provide theoretical information for its subsequent anticancer drug development. We have make a major revision to our manuscript according to your comments and other two reviewers' requests and questions. The revised manuscript was resubmitted to Molecules for further review.
Reviewer 2 Report
Ma and his colleagues presented their study on isolating lanostane triterpenoids and ergostane steroids from the Ganoderma. Twenty-six triterpenoids were isolated. Three of them were new compounds. The conformation of the three new triterpenoids was determined using spectroscopic methods. The twenty-six isolated triterpenoids were also put to examine their cytotoxicity against three cancer cells. The methodologies are properly designed and carefully performed. The obtained results are solid and convincing. This manuscript is recommended for acceptance for publication after minor revision.
1. The detail of NMR parameters for C13, HMBC and ROESY spectra should be described.
2. How the authors measured the cytotoxicity also needs to put the details, such as the treatment duration, which assay to measure the viability and IC50.
3. The cytotoxicity for the test compound in table 1 seems to be specific to cell phenotype. This phenomenon should be carefully analyzed and discussed.
Author Response
Thanks very much for your detailed comments and these comments are all valuable and very helpful for revising and improving our paper. The point-by-point responses to your comments are listed as follows.
- The detail of NMR parameters for C13, HMBC and ROESY spectra should be described.
R: Some NMR parameters for C13, HMBC and ROESY spectra were added in the revised manuscript as your suggestion.
- How the authors measured the cytotoxicity also needs to put the details, such as the treatment duration, which assay to measure the viability and IC50.
R: The related details of Bioassay of cytotoxic assay were supplemented in Part 3.4.
- The cytotoxicity for the test compound in table 1 seems to be specific to cell phenotype. This phenomenon should be carefully analyzed and discussed.
R: The related discussion about selectivity of of lanostanoids against cancer cell lines was analyzed and discussed in the revised manuscript.
Reviewer 3 Report
The manuscript entitled "Lanostane triterpenoids and ergostane steroids from the fruit-2 ing bodies of Ganoderma luteomarginatum and their cytotoxi-3 city" by Qing Yun Ma et al. Is an interesting work on the subject.
However, it seems to me that it is quite an underdeveloped manuscript for such a reputable journal with IF almost 5. With all due respect to the authors, they have to pay attention to this work a bit.
If I develop this manuscript properly, I will support it in the publication.
Below are my comments.
1. The title can be shortened a bit - but that's how the authors think.
2. The abstract tells very poorly about what is done at work - please rewrite it in its entirety.
3. The introduction, for such an extensive topic, is at least three times too short. Figure 1 is very nice but the introduction is way too short. By typing the keywords proposed by the authors into the browser, a lot of work in the search engine pops up. Please work it out properly.
4. Why green borders for tables?
5. The result and discussion section is basically a brief description of the results. Where is the discussion here? Please work it out.
6. The number of citations should rather be definitely expanded.
To sum up, I believe that thematically - when it comes to the undertaken research problem, the work is really interesting. But it is terribly unfinished. Before its publication, it needs to be carefully refined.
Author Response
Thanks very much for your detailed comments and these comments are all valuable and very helpful for revising and improving our paper. The point-by-point responses to your comments are listed as follows.
- The title can be shortened a bit - but that's how the authors think.
R: The title was shortened as shown in the revised manuscript.
- The abstract tells very poorly about what is done at work - please rewrite it in its entirety.
R: The abstract was almost rewritten and improved in the revised manuscript as your suggestion.
- The introduction, for such an extensive topic, is at least three times too short. Figure 1 is very nice but the introduction is way too short. By typing the keywords proposed by the authors into the browser, a lot of work in the search engine pops up. Please work it out properly.
R: We have reviewed and studied a large number of literatures related to topic in manuscript during our revision. The introduction was improved entirely due to our great efforts, in which one paragraph was added to describe antitumor efficacy of lanostane-type triterpenoids. The revised introduction part looks relatively full and optimized.
- Why green borders for tables?
R: The green borders were revised to black borders in tables.
- The result and discussion section is basically a brief description of the results. Where is the discussion here? Please work it out.
R: The discussion mainly appears in the last paragraph of both Parts 2.1 and 2.2, which were all improved properly in the light of related references. In the last paragraph of Part 2.1, structural types and characteristics of lanostanoids discovered in genus Ganoderma and our present species Ganoderma luteomarginatum were discussed. In the last paragraph of Part 2.2, the cytotoxic properties and its structure-cytotoxicity relationships of lanostanoids were discussed.
- The number of citations should rather be definitely expanded.
R: Eighteen references were properly added in the revised manuscript as your suggestion.
Round 2
Reviewer 1 Report
I reviewed the revised manuscript titled “Lanostane triterpenoids and ergostane steroids from Ganoderma luteomarginatum and their cytotoxicity” for your esteemed ‘MDPI Molecules’. This study is not a novelty.
The authors investigated “Lanostane triterpenoids and ergostane steroids from Ganoderma luteomarginatum and their cytotoxicity”.
Other comments
In view of this tight competitive situation, only manuscripts with highly innovative contents, originality, gain in scientific knowledge, straightforward experimental design, advanced state-of-art technical performance of experiments, and general interest for the readership can be considered.
In the present case, your manuscript represents there is no novelty for scientifically sound contribution because already published this kind of objective, please refer to below attached hyperlinks for your kind perusal. However, it did not reach a sufficiently high priority index concerning novelty and relevance to a broad scientific readership. Therefore, we feel that your work would not be accepted.
Decision:Reject
Links:
Ganoderma luteomarginatum - Search Results - PubMed (nih.gov)
https://scholar.google.com/scholar?hl=en&as_sdt=0%2C5&q=+ergostane+lanostane+Ganoderma+luteomarginatum&btnG=
Reviewer 3 Report
The authors improved the article very well, which was quite interesting anyway. I recommend publishing in the current form.